# Designing Efficient Neural Attention Systems Towards Achieving Human-level Sharp Vision

**Abdul R. A.Ghani, Alfredo Solano, Yusuke Iwasawa, Kotaro Nakayama, Yutaka Matsuo**
Graduate School of Engineering, The University of Tokyo, Japan
{abdu,asolano,iwasawa,nakayama,matsuo}@weblab.t.u-tokyo.ac.jp

**Nishanth Koganti**
Graduate School of Science and Technology, Nara Institute of Science and Technology, Japan
nishanth-k@is.naist.jp

## Abstract

Human vision is capable of focusing on subtle visual cues at high resolution by relying on a foveal view coupled with an attention mechanism. Recently, there have been several studies that proposed deep reinforcement learning based attention models. However, these studies do not explicitly consider the design of a foveal representation and its effect on an attention system is unclear. In this paper, we investigate the effect of using a hierarchy of visual streams in training an efficient attention model towards achieving a human-level sharp vision. We perform our evaluation on a simulated human-robot interaction task where the agent attends to faces that are looking at it. The experimental results show that the performance of the system relies on factors such as the number of visual streams, their relative field-of-view and we demonstrate that maintaining a hierarchy within the visual streams is crucial to learn attention strategies.

## 1 Introduction

Humans perform tasks such as driving or engage in social interactions by paying attention to subtle visual cues such as the intention of a pedestrian or facial expressions with very high acuity (1/120 degrees Fahle & Poggio (1981)). This is achieved by a foveal view, consisting of a hierarchy of visual representations, coupled with an attention mechanism Pickrell (2003) as shown in Figure 1a. This hierarchy plays a crucial role for effective attention mechanisms as the wider representations contain low-resolution information of the environment which is used to guide the narrower representations in a hierarchical manner towards the regions-of-interests (ROIs).

To reach all-around human-level sharpness a traditional camera has to capture images in the order of $5 \times 10^8$ pixels Fahle & Poggio (1981), which can't be realized at the present time (and in the foreseeable future). Due to such hardware limitations, existing systems perform poorly in real-world environments Rajaram et al. (2016); Dodge & Karam (2016); Siam et al. (2017). To tackle this problem, there have been early attempts to replicate human vision with hierarchical visual streams

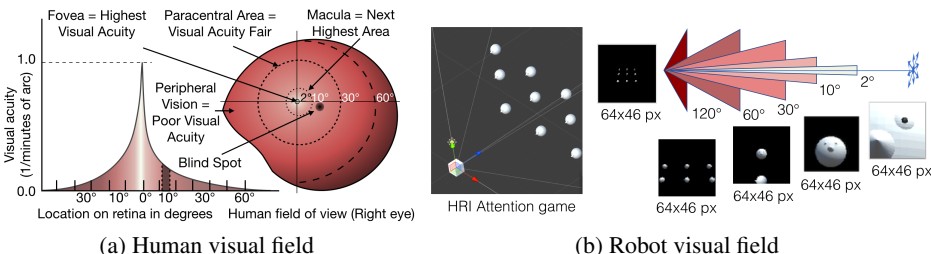

(a) Human visual field        (b) Robot visual field

Figure 1: Comparison between (a) Human visual field and (b) proposed Robotic visual field.

for object recognition Bandera et al. (1996). These studies evaluated the performance in controlled settings and relied on hand-crafted features. On the other hand, recent studies have proposed scalable attention models and performed end-to-end training using deep reinforcement learning Mnih et al. (2014). However, these studies do not explicitly discuss the design of a foveal view and the effect of such hierarchy on the efficiency of a deep reinforcement learning based attention model is unclear.

In this study, we investigate the effect of using a hierarchy of visual streams in training a deep reinforcement learning based attention model. We perform the evaluation in two settings 1) a simulated human-robot interaction task that relies on the detection of local features (eyes) so that the agent can learn to attend to faces that are looking at it as shown in Figure 1b, 2) classification of hand-written digits similar to the task setting in Mnih et al. (2014). We explore several factors such as the number of streams, relative angle of views and resolutions. The experimental results demonstrate that hierarchical visual streams are crucial to learning control strategies with subtle visual cues. Our long-term goal is to use such a hierarchy in realizing a human-level sharp vision that can be used in real-world applications such as human-robot interaction (HRI) where the agent can attend to subtle visual cues (Figure 3).

## 2 METHODS

To reach our goal of human-level sharp vision, we need to evaluate the importance of a foveal view in an attention model. In this study, we investigate three factors that could affect the performance for hierarchical attention: 1) field of view (FOV) for an image stream, 2) hierarchy between image streams and 3) number of image streams. We designed 8 representations to evaluate these factors as shown in Figure 2. For experiment 1, we consider a single camera with different FOV. For experiment 2, we consider two cameras with different combinations of FOV and experiment 3 has an increasing number of cameras. As a baseline, we also consider a fully-observable setting where the environment state is provided to the network.

Human-robot interaction is a challenging task that requires the agent to detect subtle visual cues such as eye gaze. We designed a simulated HRI task as shown in Figure 1b to perform our evaluation. The environment has nine toy 3D-faces on a 3x3 unit grid. The agent consists of a set of cameras where each camera captures an image stream at a different FOV. In each episode, a randomly selected face will look toward the agent while the other faces will look in random directions. The task for the agent is to locate the face that is looking at it. At each time step, the agent can rotate the cameras by one degree in any direction. The reward function includes two components: at each time step the agent receive a punishment of -0.1 and a terminal reward of +1 if it can locate the correct face. Each episode has maximum steps of 50.

To train the attention model, we use Proximal Policy Optimization (PPO) Schulman et al. (2017) which is an efficient policy search algorithm capable of handling high-dimensional observations. The policy function is represented by a DNN as shown in Figure 4. The input to the network includes both the environment state and observations. The state representation is given by $s_t \in \mathbb{R}^{2n_c}$ with $n_c$ as the number of cameras and the state for each camera as its current orientation along two axes of rotation. Each camera also provides an observation in the form of an RGB image $o_t^{(c_i)} \in \mathbb{R}^{64 \times 64 \times 3}$. There are two outputs generated by the network which includes the policy and a scalar state value estimate. The policy output is a softmax probability distribution $\pi_\theta(a_t|s_t) \in \mathbb{R}^9$ where the actions include the motion of the cameras along one of the eight cardinal directions and no action.

To further validate the effect of a hierarchical representation, we consider a hand-written digit classification task using the recurrent attention model proposed by Mnih et al. Mnih et al. (2014). For this classification task, the agent would need to summarize the whole image efficiently rather than focus on sharp features required in the HRI task. The size of an ROI used with the attention model was varied based on the representations shown in Figure 2. The mapping from FOV to image windows sizes was computed by relying on coordinate transformations w.r.t a fixed viewpoint.

## 3 RESULTS AND DISCUSSION

In this section, we present our evaluation on two task settings. For each task setting, we trained an agent with each of the foveal representations shown in Figure 2 until convergence. To ensure

generalizability, we repeated the experiments with three different seeds and the learning curves shown in Figure 2 are the average over these runs. The performance of all the experiments are summarized in Table 1. The values were computed by the average score over the last 100 episodes. It can be seen that there are large variations in performance depending on the foveal representation used and it is a crucial hyperparameter for an attention model.

From Figure 2a, we show that the camera with FOV of $30°$ is the best representation for this game as a single observation, while cameras with other FOV ($10°$, $60°$) fails. This indicates that with a single observation, the FOV can greatly affect performance. From Figure 2b experiments 5,6 have low performance as the combination of two observations have large variations in FOV and the wider camera is unable to guide the narrower camera. On the other hand, experiment 7 has the best performance as both cameras are able to work together efficiently which indicates forming a hierarchy is another crucial aspect. Finally from Figure 2c, we can notice that increasing the number of cameras has increasing performance as they capture information from the environment at an increasing number of resolutions hierarchy.

We observe similar performance with the image classification task as shown in Table 1 with experiment 8 having the best performance in both cases. However, there were slight variations in comparison to the HRI task such as experiment 4 being better than experiment 3. The reason for such variations could be that the agent needs to focus on global images features and works better with wider FOV in general, similar observation can be made from experiment 6 and 7.

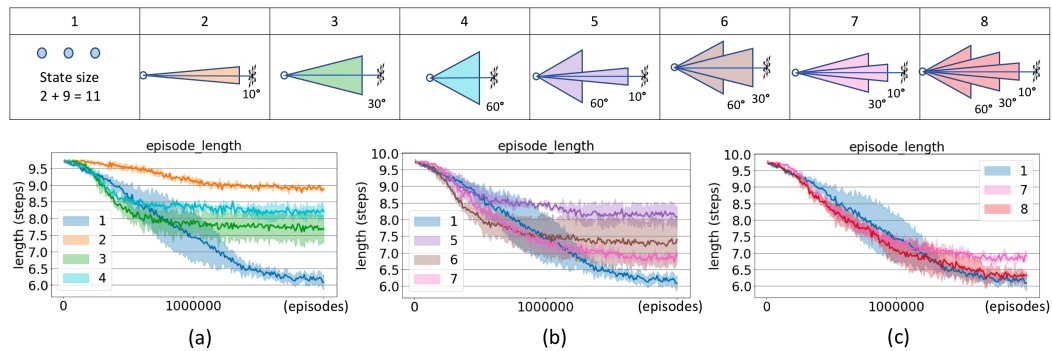

Figure 2: Top: Foveal representations used for evaluation. Bottom: Comparison of episode length for different experiments.

Table 1: Cumulative Reward for HRI Game and Digit Classification

| Experiments | 1 | 2 | 3 | 4 | 5 | 6 | 7 | 8 |
|---|---|---|---|---|---|---|---|---|
| HRI Game (PPO-Attention) | 0.204 | -0.73 | **-0.47** | -0.60 | -0.59 | -0.25 | **-0.16** | **0.04** |
| MNIST (RAM) | - | 0.680 | 0.900 | **0.935** | 0.905 | **0.950** | 0.865 | **0.970** |

## 4  CONCLUSION

Sharp vision is crucial to achieve proficiency in many real-world applications. Recently, deep learning based attention models have shown promising results towards this direction. However, they do not scale to achieve human-level sharpness. In this study, we investigate the effect of a hierarchical representation in training an attention model. The experimental results demonstrate several factors that need to be considered for designing an efficient attention model. Our future work is to rely on such insights to achieve a human-level sharp vision for real-world applications. We are also working on a hardware implementation of such hierarchical representation on a mobile robot (Figure 3) suitable for applications such as human-robot interaction.

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

## APPENDIX A   ROBOT, SIMULATOR AND THE NETWORK

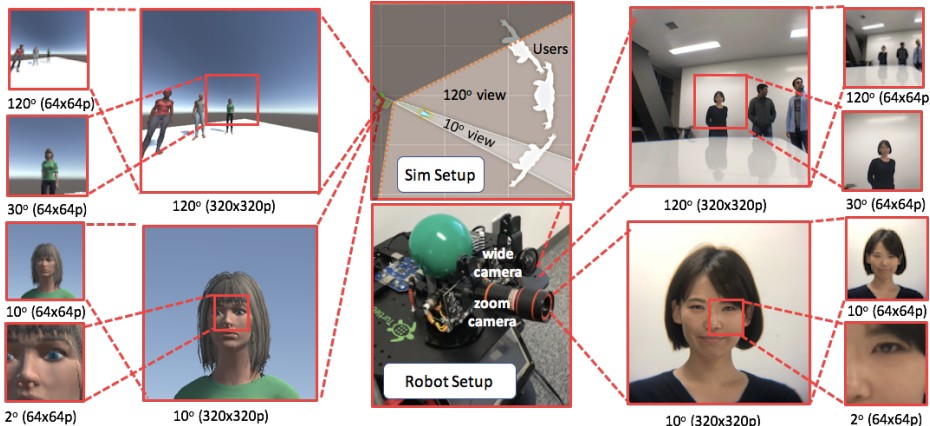

Figure 3: Robot and simulator setups, we used two cameras (with wide and focused lenses), and generated four hierarchical streams as input for the network, the FOV of each stream is been selected to help in HRI task. where the widest stream with $120°$ can locate the person to attend to, the stream with $30°$ can locate parts of the human body such as a face, the stream with $10°$ can direct the sharpest stream to the region of interests for inference ROI.

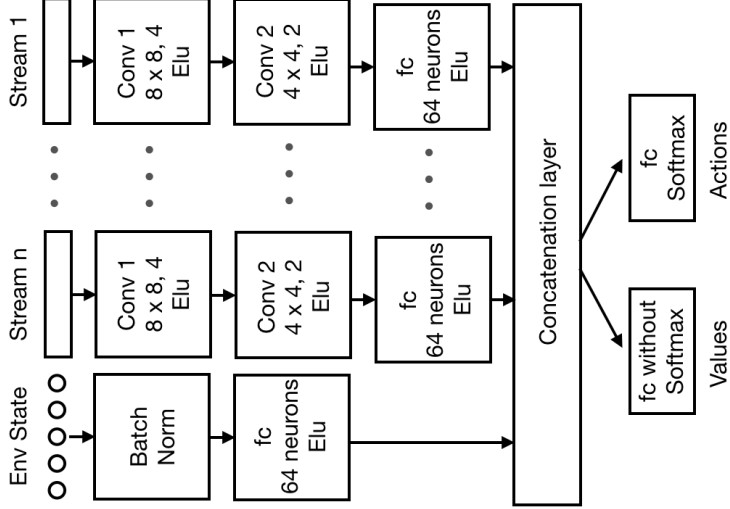

Figure 4: Neural Network graph, all network streams are concatenated on the last fully connected layer before the values and actions layers, this type if concatenation is been studied in Karpathy et al. (2014).

