# OpenReview forum: "Designing Efficient Neural Attention Systems Towards Achieving Human-level Sharp Vision"
_ICLR.cc/2018/Workshop — Accept_

### Official Review · AnonReviewer3 · 2018-03-07
**Somewhat interesting / overlooked area of research without clear / salient results**

**Rating:** 5
**Confidence:** 4

**Review:**

This paper investigates strategies for combining multiple input streams from different cameras with different FOVs using a deep reinforcement learning based approach. The approach is relatively straightforward but the specific problem has not been studied extensively. The evaluation using a simulated human-robot interaction task is interesting but as described seems quite narrow (and constrained). It is also probably not representative of real-world human-robot interactions where many image nuisances not modeled here (e.g., illumination, face variations, distance, clutter, etc) contribute to making the task challenging. The evaluation does not provide any compelling case for such foveating system to outperform a standard non-foveating one. It is also unlikely that the specific results regarding the best combination of FOVs would generalize to other problems.

---

### Official Review · AnonReviewer1 · 2018-03-10
**I think the paper is useful and relevant and recommend acceptance.**

**Rating:** 7
**Confidence:** 4

**Review:**

The paper studies visual stream parameters for detailed perception and its impact on localisation performance.

I think the paper is very well written, concise and contains all the relevant details for the experiments. The paper seems original and can be very useful as we start developing more and more complex robots that will do more complex tasks.

pros:
- simple and clear goals and experimentation
- clearly a step in an important direction

cons:
- the scope is a bit narrow but I find it useful nonetheless.

---

### Decision · Program_Chairs · 2018-03-20
**ICLR 2018 Workshop Acceptance Decision**

**Decision:**

Accept

**Comment:**

Congratulations, your paper was accepted to the ICLR workshop.